# Semantic Digital Twins: security and scale for constrained IoT devices

*Evandro Pioli Moro[1] and Alistair K Duke[2]

[1] p/p 9, 5th floor, Orion Building, Adastral Park,
Ipswich, IP5 3RE, United Kingdom

[1]*evandro.moro@bt.com
[2]alistair.duke@bt.com

* Corresponding Author

**Abstract.** Digital twins are largely used to enable the Industry 4.0 and other Internet of Things applications. This approach has been built to support the product entire lifecycle and has advantages such as interoperability, availability, and simulation capabilities. If integrated with distributed ledger technologies, such as blockchains, digital twins can augment their applications, becoming an anchor of trust for constrained IoT devices. This paper describes the Constrained Devices Management Platform, a BT research system designed to enhance the security of IoT systems, with an especial focus on data provenance, immutability, and integrity. Lastly, the paper describes how semantics can enhance the CDMP capabilities through interoperability.

**Keywords:** Digital twins, internet of things, distributed ledger technologies, blockchain, cybersecurity, ontology, interoperability.

## 1    Introduction

Digital Twinning can be described as a method for turning physical devices into digital services [1]. These services can be used to satisfy the interaction requirements placed on physical devices in a more efficient and scalable way [2]. Application Programming Interfaces (APIs) allow control and access to the digital assets either by direct communication or to replicate this behaviour if the devices are not online or only send data periodically [3]. The APIs typically support requests such as 'What is the latest data?' and 'What are the changes to the state of the device?'. It is also possible to enhance the capability of the APIs with value-added data from elsewhere [4], semantic operations [5], or via analysis [6].

The approach has been developed to support the lifecycle of devices throughout

design, manufacture, and operation [7]. During the design process, the APIs can be used in simulations before the device itself even exists. During manufacture, the results from the simulations can be used to enhance and optimise the manufacturing processes. While during operations, the digital twins (DTs) can provide a virtual simulation environment and optimise data operations, reducing the communications requirements, hence reducing the latency and increasing battery life.

One of the most prominent applications of DTs is to support the enablement of the Industry 4.0 [8]. Industry 4.0 provides data operations to devices and products throughout their entire lifecycle—from concept to recycling [9]. It is often branded as the Industrial Internet of Things (IIoT). This data can be used to influence the manufacture of the product, for example, or to provide real-time lifecycle management to an entire line of products. DTs are centric to this approach, as it enables a virtual entity of the physical devices so that other processes can provide insight and actuation to the entire system.

## 2 Internet of Things

### 2.1 Introduction

The Internet of Things (IoT) has various meanings in different applications. However, one standard definition that is central to the concept of DT is the provision of connectivity and data exchanging facilities to various devices for real-time data operations [10].

The rapid deployment of IoT systems has delivered exceptional opportunities to various industries, especially regarding data capture and observability. The cost of deployment of IoT has been decreasing over time, but the attention to cybersecurity has not accompanied the pace of IoT developments, once it is often seen as an economic harness to the solution [11]. Devices are getting cheaper and more computational constrained, which makes running traditional cybersecurity protocols on them unfeasible.

### 2.2 Relevance of DT for IoT cybersecurity

Centric to the research that British Telecommunications (BT) do on IoT systems is applying breakthrough networking technologies to enhance the cybersecurity of constrained IoT devices. Recently, the combination of DTs and blockchain has proven useful to counteract cyber threats. DTs provide four main advantages to support cybersecurity applications at a networking level:

- Simulation environment. By using real-time data, the DT may incorporate predictive analytics so that it can help to determine when an asset is likely to fail and what are the potential consequences of such fault. Alternatively, predictive analytics can be used to optimise the overall system operations [6].

- Data sharing across siloes. DT can act as a data proxy to collect data from sensors in different environments, instead of each silo capturing their own data, enhancing interoperability [12].

- Real-time operations management. The digital assets create situational awareness by building a holistic picture of the environment it represents. By enabling real-time visibility, recommendations and actions, the users can perform real-time operations using decisions based on data [3].

- Availability and reliability. Because DT does not rely on the availability of communication channels, DT provides a globally shared understanding of what an entity looks like in near real-time [3].

## 2.2 Relevance of Distributed Ledger Technologies for IoT cybersecurity

Distributed Ledger Technologies (DLT) have their most prominent application on blockchains. DLT is a peer-to-peer networking system where a ledger of transactions is synchronised among peers [13]. If the transactions ledger is immutable and structured in the form of blocks of information linked to the previous blocks via a hashing operation, then the DLT is called a blockchain. The transactions synchronisation process occurs via a protocol called consensus. The consensus protocol is often a computationally-intensive process [14] to chose the peer responsible for generating the new block by validating and aggregating the transactions. Consensus protocols also set how long it takes for a block to be generated (and consequently, how long it takes for the transactions to be validated). Various consensus protocols exist, and their choice is paramount to the success of the blockchain [15].

Blockchain networks offer various advantages when compared to traditional distributed data sharing facilities (including databases). These include immutability, enhanced fault tolerance, auditability, transparency, and decentralised trust provision [16]. However, blockchains might also offer drawbacks such as increased computational power consumption and longer transaction processing times [14] (in 2020, the public Bitcoin network took between 8-11 minutes to validate a transaction by creating a new block [17]).

# 3 Constrained Devices Management Platform

Combining the aforementioned cybersecurity advantages of DT with the unique trustless and immutability features of blockchains, BT introduced a prototype platform called Constrained Devices Management Platform (CDMP).

With a particular focus on data integrity and data provenance for battery and network-constrained IoT devices, the CDMP reduces the networking requirements of the IoT sensors by performing the heavy security operations on a cloud service. The cloud operations include data integrity, provenance, and immutability by combining sensors digital twinning with blockchain storage and hashing operations.

The data from a sensor is transferred in a local area network (LAN) to a network gateway. Upon arrival at the gateway, the message is signed using blockchain scripts (smart contracts) where the message identifier (msgID) and the message hash (msgHash), along with other use case-specific parameters, are stored within the blockchain, and hence are immutable. The gateway will then update the DT with the latest data available. Upon retrieval, the end-user can verify the integrity and provenance of the IoT data fetched from the DT by retrieving the msgHash from the blockchain and comparing it with the hashing operation performed locally. The protocols/APIs used to exchange data with the CDMP are secured on the enhanced gateways, where outgoing communications are completed over a virtual private network (VPN).

The DT framework utilised on this technology concept was Eclipse Ditto. Ditto was integrated with a private instance of the Ethereum blockchain framework, set with a proof-of-authority (POA) consensus algorithm. With POA, the transactions could be processed quicker (in less than a second) without compromising security.

The CDMP, therefore, offers the end-users the ability to perform digital twinning of physical assets while being able to verify the integrity of the messages via crypto operations. The diagram in Fig. 1. depicts the data flow on a generic IoT environment where the CDMP is used.

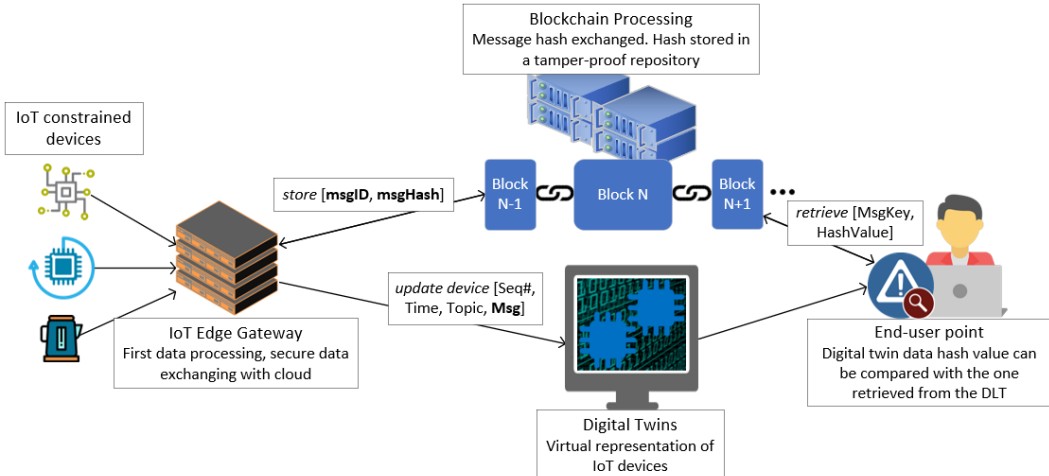

**Fig. 1.** Constrained Devices Management Platform basic components architecture and data flow

## 4 Semantics in a Digital Twins for cybersecurity context

In the current platform, devices are reflected in the DT by a set of attributes and features. The attributes represent the more static properties of the device. The features, which might include a set of properties and a set of desired properties, are used to model the more dynamic aspects of the physical device. Because this framework favours flexibility, allowing the DTs to be very closely aligned with the physical asset, various data models might co-exist in the virtual world. This makes it challenging to manage a system involving many device types where data might be defined in different and even contradictory ways. From a cybersecurity perspective, identifying problematic devices or messages using a collective threat model becomes difficult, potentially requiring bespoke translation to each type of device or message, increasing the complexity of such a system.

Adopting an approach based on a common ontology will help to solve these issues. One such approach would be to introduce an abstraction layer between the different devices and the DT. One example is the one employed by Eclipse Vorto[1]. Vorto allows the creation and management of technology agnostic, abstract device

---

[1] http://www.eclipse.org/vorto/

descriptions. An alternative approach would be to create a relationship between the more device-specific twins and a generic ontological representation that forms the abstraction layer. Here, the twin features would contain metadata creating a mapping to an ontology describing the device type or data type concerned. This way, the device data can be expressed in terms of one or more ontologies, facilitating querying and reasoning over the data at the ontological level. Both approaches would allow threat models, which describe suspicious behaviour associated with particular types of devices, to be evaluated in systems involving multiple instantiations of those device types (e.g., different manufacturers or versions).

## 5 Conclusion

This paper explored the use cases of semantic digital twins for light-weight cybersecurity, focusing on battery and network-constrained IoT devices. It discussed the Constrained Devices Management Platform (CDMP), a system leveraging the combination of DT and blockchain to provide data integrity and data provenance for battery and network-constrained IoT devices. The CDMP improves the network operations and security of IoT devices, reducing the battery and computational power requirements. The paper then discussed the impact of the semantics in DT and how interoperability is vital in an IoT cybersecurity scenario.

The next steps for the CDMP development include customer trials scheduled to start in May 2021. The DMP will incorporate the systems of the ongoing project i-Trace[2], and Innovate UK project with Costain, Cisco, the University of Warwick, and Senseon. i-Trace is an end-to-end cybersecurity solution using DT, blockchain, and artificial intelligence to provide a resilient security framework to IoT devices. The use cases will likely be related to smart construction sites. The CDMP will compose the overall architecture to provide the data integrity and data provenance facilities.

---

[2] https://www.i-trace.co.uk/

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
