# OpenReview forum: "Semantic Digital Twins: security and scale for constrained IoT devices"
_eswc-conferences.org/ESWC/2021/Workshop/SeDiT — SeDiT 2021 Oral_

### Official Review · AnonReviewer2 · 2021-03-28
**An Edge-Platform for IoT Data**

**Rating:** 6
**Confidence:** 4

**Review:**

The authors present a Distributed Ledger approach for IoT data. The integrity and immutability of data streams is ensured at the edge of the networks, thereby releasing the – resource-constraint – data sources from the Distributed Ledger operations.
The paper seems to describe a commercial product in a relatively early stage. That makes the motivation and already gained insights interesting for the community, and worth to discuss at the workshop. There are however a few shortcomings, which do not help the reader to understand the intended message.

At the core, the paper claims to support cyber-security by signing data streams and storing the proofs in a Blockchain. The argumentation however often mixes Digital Twin concepts from simulation, interoperability, semantics etc. Focusing on the one main contribution and explaining that in more details will make the paper much stronger.

The claimed advantages of the proposed platform are also hardly justified in the rest of the paper. That’s somehow understandable as the platform itself is still under development. Nevertheless, more insights why BT itself, its partners and customers want to invest into the platform and how it intends to meet the requirements are certainly important for the community.

To put everything together, the added contribution of the platform above the state of the art seems rather limited, however the insights and motivations for implementing it in a commercial setting are highly relevant for the community discussion. If the authors can focus more on these aspects, the presentation at the workshop can be justified.

Ideas for improving the underlying DT concepts:
* The W3C Web of Things activities, the Asset Administration Shell and the Industrial Internet Consortium (and many others) promote standardized data models, also in form of ontologies. Using them in the platform can directly increase the reusability of the modeled DTs and help with the interoperability with other platforms.
* Distributed IDentifier (DID) and Verifiable Credentials (VC) sound like a well-fitting extension to the proposed platform. They are broadly known standards and can be connected with Distributed Ledgers.
* Distributed Ledgers are always expensive, and a practical solution needs to state a model who and why shall cover these costs. For instance, if only a small ratio of the participating parties has a motivation to compute the blocks, a centralized system might be more efficient. Elaborating ideas here will make the platform way more convincing.


The paper uses abbreviations before introducing them, for instance in the abstract (BT, CDMP even though the long term already appeared).

2.1 “Devices are getting … more computational constrained, …” --> It’s rather the other way around, previously not connected devices become internet-ready, therefore the computational abilities raise.

The topics mentioned in the bullet points are correct, however they are independent of (cyber-)security. It is therefore not clear why the authors see DTs as a tool to increase the security level.


The text has a few grammar errors, just three examples:

Abstract: …support the *entire* product lifecycle…

2.2 that British Telecommunications (BT) *does* on IoT systems

*2.3* Relevance of Distributed Ledger Technologies for IoT cybersecurity

---

### Official Review · AnonReviewer1 · 2021-03-28
**A platform in early development stages, has good potential**

**Rating:** 6
**Confidence:** 4

**Review:**

This paper describes the Constrained Devices Management Platform, a platform that enables IoT devices with limited computational resources to store data within a Digital Twin as well as Blockchain for increased security.

The architecture of the platform has its merits as it shifts the security and analytics away from the IoT devices. The use of Blockchain adds a layer of security ensuring that stored data is valid.

However, the role of Digital Twins in the architecture is rather limited. The Digital Twin serves as a database of latest data from the device, while Blockchain is used in order to verify the validity of data retrieved from the Digital Twin.

The cybersecurity perspective is not elaborated in terms of how problematic devices or messages will be identified or dealt with. Data integration on an ontological level, as well as querying and reasoning, is currently future work.

---

### Official Review · AnonReviewer4 · 2021-04-07
**Semantic Digital Twins: security and scale for constrained IoT devices**

**Rating:** 6
**Confidence:** 4

**Review:**

The paper presents a platform to improve the security of IoT data shared across a DT platform by relying on blockchain technologies. The objective is to liberate constrained IoT devices from expensive security processes and transferring them to dedicated cloud services, without compromising the security and integrity of the data.

Strong points:
* Security is a relevant aspect of DT and IoT that must be addressed.
* The architecture of the system is clear to understand.
* The use of blockchain technologies is an interesting approach in order to ensure data integrity within a DT framework.

Weak points:
* In section 2: It is unclear how DT capabilities such as simulation, data sharing, real-time, etc can support cybersecurity applications. From my understanding, the proposed CDMP Platform works in an isolated way, without interacting with the DT.
* Currently the DT platform is only used as a data repository with multiple data models for each device.
* it is not clear how is the procedure to deal with corrupted data if the hashes do not fit. Which is the role of the CDMP and DT platforms in that situation.

---

### Decision · Program_Chairs · 2021-04-08

Accept (Oral)